# Forensic Anthropology as a Discipline

**DOI:** 10.3390/biology10080691

**Published:** 2021-07-21

**Authors:** Nicholas V. Passalacqua, Marin A. Pilloud, Derek Congram

**Affiliations:** 1Department of Anthropology and Sociology, Western Carolina University, Cullowhee, NC 28723, USA; 2Department of Anthropology, University of Nevada, Reno, NV 89557, USA; mpilloud@unr.edu; 3Department of Archaeology, Simon Fraser University, Burnaby, BC V5A 1S6, Canada; d_congram@sfu.ca

**Keywords:** forensic anthropology, bioarchaeology, qualifications, expertise, knowledge, ethics, education, professionalization, standards

## Abstract

**Simple Summary:**

Forensic anthropology in the United States is a specialization within the overall field of anthropology. Forensic anthropologists are specially educated and trained to search, recover, and examine human remains within a medicolegal context. Over time, forensic anthropology has become increasingly specialized and distinct from other specializations within anthropology. As such, we argue that forensic anthropology should be considered its own discipline, with a unique knowledge base, separate from other similar forms of anthropology, such a bioarchaeology. We argue that forensic anthropologists have unique expertise, making them the only type of anthropologist qualified to perform medicolegal examinations of human remains. Finally, we contend that to perform or represent yourself as a forensic anthropologist without the appropriate expertise is ethical misconduct. The value of this paper is that it explains the importance of expertise and knowledge, and how forensic anthropology has diverged from other specializations of anthropology enough to be considered its own discipline.

**Abstract:**

This paper explores the current state of forensic anthropology in the United States as a distinct discipline. Forensic anthropology has become increasingly specialized and the need for strengthened professionalization is becoming paramount. This includes a need for clearly defined qualifications, training, standards of practice, certification processes, and ethical guidelines. Within this discussion, the concept of *expertise* is explored in relation to professionalization and practice, as both bioarchaeology and forensic anthropology have different areas of specialist knowledge, and therefore unique *expertise*. As working outside one’s area of expertise is an ethical violation, it is important for professional organizations to outline requisite qualifications, develop standards and best practice guidelines, and enforce robust preventive ethical codes in order to serve both their professional members and relevant stakeholders.

## 1. Introduction

Bioarchaeology and forensic anthropology are two closely related specializations of biological anthropology that examine human remains to understand the life experience and biological parameters of the individuals from which the remains are derived. In this treatment, we focus specifically on these two disciplines as they are closely linked by their study of anatomically modern human skeletal remains. We also limit our focus to the United States, recognizing that there are distinct education, practice, and professional qualification standards in different countries; in part stemming from different national/regional education systems and legal statutes. While forensic anthropology and bioarchaeology have different goals, both disciplines use similar approaches and sometimes the same methods to examine human remains, typically, gross skeletal material (to include bones and teeth) to determine such parameters as species (to ensure the remains are human in origin), sex (sometimes gender in conjunction with other contextual information), age (sometimes stage-of-life/life history), and stature (living height). Both disciplines also perform more complex analyses examining such characteristics as population variation in terms of biological distance (sometimes called ancestry/population affinity/boaffinity), antemortem and perimortem trauma, taphonomic modifications (sometimes postmortem interval), anomalous and pathological conditions, skeletal indicators of biological stress, and inferential data using archaeological context (sometimes mortuary patterns).

Bioarchaeology and forensic anthropology derive from biological anthropology, but are, at least in their ideal forms as practiced in the model of the United States, hybrids of both biological anthropology *and* archaeology. Both disciplines require the understanding of human bone biology *as well as* archaeological context and taphonomic changes to generate comprehensive conclusions about the lives (and in some cases the death or death event) of individuals. They draw from agency theory regarding the introduction of human remains into the archaeological record usually via culturally intentional actions for various purposes that can reflect culture and cultural identity more broadly [1]. Additionally, both are firmly entrenched in anthropology through their biocultural approach to understanding human biological adaptation, or the interpretation of skeletal modifications (during life, at death, and after death) through a cultural lens. 

In the current paradigm, it is not uncommon for an individual trained in one subdiscipline of biological anthropology to offer expertise and services in another, and this is particularly common between bioarchaeology and forensic anthropology. In fact, Ubelaker [2] p. 137, claimed, “[t]he symbiotic and dynamic relationship of these academic areas greatly improves the quality of the applications of each”. Contrarily, Juarez [3] argued that a focus on the commonalities between bioarchaeology and forensic anthropology is problematic as it does not emphasize the differences and boundaries of each discipline. Thompson [4] p. 68, agrees that viewing the work of a forensic anthropologist as being easily done by any trained osteologist is “a misperception of what the subject involves through focusing on methods while ignoring context”. While, Ross [5] argued that forensic anthropologists are inherently more stringent in analyses and could do all that a bioarchaeologist can do, but not vice versa. While we agree with Ubelaker [2] in that both disciplines benefit from each other, we also agree in *concept* with Juarez, Thompson, and Ross in that both disciplines are becoming increasingly complex and specialized, such that education and training in one discipline do not translate into competency or expertise in the other.

This disagreement has precipitated the need for increased professionalization in terms of standardizing education, defining qualifications, defining and implementing ethical codes, and reconsidering the roles played by professional organizations within both bioarchaeology and forensic anthropology. We view all these issues as interwoven and each influencing the other; however, thus far, they have not been explicitly addressed comprehensively in the literature. 

As both bioarchaeology and forensic anthropology have grown significantly in the last few decades, it has become prudent to explore their differences and similarities and the need for their individualization and professionalization in terms of defining qualifications (i.e., education and training needed to demonstrate adequate knowledge to perform discipline-related tasks in an applied setting) and expertise. These topics must also be framed as issues that would best be addressed by professional organizations, as disciplinary leaders harnessing the power of their communities of practitioners. This exercise is not a means of *academic gatekeeping*; but rather a means to identify minimum standards and best practices of what to expect *at a minimum* from an individual practicing a particular profession [6,7]. 

The goal of this paper is to consider both bioarchaeology and forensic anthropology as unique disciplines, having diverged due to increasing specialization and scholarly distancing; thus, bioarchaeologists and forensic anthropologists have their own unique areas of expertise and spheres of practice. While bioarchaeology and forensic anthropology can vary greatly in their education and practice globally, we focus on the practice of these subdisciplines within the United States. For overviews of forensic anthropology in other countries, there are several excellent treatments to which the reader can refer, e.g., [8,9,10,11,12,13,14,15,16,17,18,19,20,21,22,23,24,25,26,27,28]. In this treatment, we begin with a brief discussion on the trend of increasing specialization and decreasing overlap in educational programs and scholarship between the two disciplines. We follow with a definition and discussion on the scholarship of expertise and its relevance to considering bioarchaeology and forensic anthropology as unique expertise. We then expand this discussion within the context of professional qualifications, primarily in regard to the role of ethical codes and professional organizations. Next, we provide an overview of qualifications and their importance in relation to expertise, education, and practice, followed by a discussion of how to codify expertise and practice using best practice recommendations and standards documents, which are becoming ever more popular within the forensic sciences and may soon be required for practice within that context. Finally, we conclude with recommendations for the future, a call for greater consideration of the importance of qualifications as a means of respecting both the remains of those we study and their extant next-of-kin/communities, and provide a glossary (Glossary) defining several of the terms used throughout this paper for standardization and clarification.

### The Divergence of Bioarchaeology and Forensic Anthropology

Early versions of both bioarchaeology and forensic anthropology were originally practiced by physicians, anatomists, and biological anthropologists with interests in the examination of the human skeleton. The examination of skeletal remains, as well as the types of research questions addressed, have always been dependent upon the contexts from which the remains were derived. When skeletons within archaeological contexts are excavated, researchers want to know about the life experiences of these individuals. Hypotheses may be formulated to pursue research around migration, diet, stress, violence, social structure, disease loads, activity levels, disability, mortuary practices, fertility, demography, growth and development, and life history, among many others, e.g., [29,30,31,32,33,34,35,36,37,38,39,40,41].

When modern skeletons are discovered, anthropologists and the medicolegal community want to know the identity and circumstances of the death of that individual. To pursue identification, they may estimate the individual’s biological profile (i.e., age, ancestry/population affinity, sex, and stature), describe individualizing features, and compare ante- and postmortem data, e.g., [42,43,44,45,46,47,48,49,50,51,52,53]. They are also interested in the circumstances surrounding the death event, illustrated by perimortem trauma and taphonomic alterations, and potentially estimating a postmortem interval, e.g., [54,55,56,57]. Research also exists on the applicability of indicators of biological stress as part of the identification process [58] and investigations into gross human rights violations and structural violence [59,60,61]. However, the collection or analysis of such data is not routinely performed as part of forensic anthropological casework (i.e., reports provided in a medicolegal context). 

Over the past several decades, various methods have been developed to best address the research agendas of each discipline, with differing foci based on the context-dependent nature of these investigations. Academically, these differing research agendas have increasingly diverged into academic programs and graduate advisors specializing in bioarchaeological *or* forensic anthropological approaches. In doing so, bioarchaeology and forensic anthropology have slowly deviated in terms of professional conferences attended [62], academic advisors and institutions, bodies of literature, venues of publication, and professional memberships. Through this divergence, they have become more and more isolated from each other, developing separate communities of practice with separate “social lives” [63]. All scientific disciplines essentially function in this manner, effectively “mold[ing] their disciplines by pedagogically fashioning their disciples” [64] p. 381. The choices made by academic hiring committees for future directions of a program are the same as those made by graduate student advisors in that they are purposeful, active choices, which intentionally shape future generations of pedagogy [64,65,66,67,68].

It is unclear precisely when individuals in the United States studying human skeletons from archaeological contexts, being educated in biological anthropology, began to identify as *bioarchaeologists*, or when individuals studying human skeletons in medicolegal contexts being educated in biological anthropology began to identify as *forensic anthropologists*; that is, as opposed to identifying as biological/physical anthropologists. It is likely that as each subdiscipline increased in popularity, practitioners began to self-identify as one or the other. According to Snow [69] and Tersigni-Tarrant and Langley [70], individuals began identifying as forensic anthropologists in the 1970s; however, this trend grew in the 1980s when forensic anthropology began to gain popularity from the work of William Bass, Walter Birkby, William Maples, and their graduate students (John Williams, personal communication 2019). 

In the 1990s, bioarchaeology began to increase in visibility with the passage of the Native American Graves Protection and Repatriation Act (NAGPRA), which required trained osteologists to assist in repatriation efforts. Additionally in the 1990s, two formative publications in bioarchaeology were released: *Standards for Data Collection from Human Skeletal Remains* [71] and Clark Spencer Larsen’s [72] *B**ioarchaeology: Interpreting Behavior from the Human Skeleton*. It is likely, then, that around the 1980s and 1990s individuals began to identify as either bioarchaeologists or forensic anthropologists, pursuing graduate advisors based on such foci and graduate programs with discipline-specific education programs. While there are some individuals who practice both and who consider themselves a bioarchaeologist and a forensic anthropologist, this number has likely decreased over the past several decades based on our observation of the subdisciplines, and may continue to do so as each becomes more specialized. An increase in full-time applied positions for forensic anthropologists [73] has also surely influenced this trend.

Considering the divergence of bioarchaeology and forensic anthropology, Buikstra et al. [74] found large increases in publications focusing on bioarchaeology *and* forensic anthropology starting in the 1970s, corresponding with the incipient formalization of both areas of study. However, Buikstra et al. [74] also demonstrated that while bioarchaeological literature was found in a variety of anthropological journals, forensic anthropological literature was increasingly published in the *Journal of Forensic Sciences* to the exclusion of other more anthropologically focused journals. Bethard [75] also demonstrated this trend by practicing forensic anthropologists certified by the American Board of Forensic Anthropology (ABFA). Bethard [75] found that based solely on the focus of dissertation subjects, representing the focus of graduate research projects, forensic anthropologists have increasingly pursued forensic anthropological research topics, rather than bioarchaeological or other more general biological anthropology topics, particularly since 2005.

Only recently have the first journals dedicated to bioarchaeology or forensic anthropology been established. Arguably, the first journal dedicated to bioarchaeology was the *International Journal of Osteoarchaeology*, established in 1991 and published by Wiley. Although, like the term osteoarchaeology itself, this journal has a heavy European focus and includes many publications on the analysis of non-human remains. It was not until 2017 that the journal *Bioarchaeology International* was established, published by the University of Florida Press [76]; this journal was followed one year later in 2018 by the first journal dedicated to forensic anthropology, *Forensic Anthropology* [77], also published by University of Florida Press (Gainesville, FL, USA). 

Martin [78] p. 163, points out that bioarchaeologists have long been critical of forensic anthropological work as being “merely technical expertise”. She fully admits that she used to be one of those bioarchaeologists who questioned “where is the anthropology in forensic anthropology?” [78] p. 163; yet, she has changed her viewpoint on the topic in a recognition that forensic anthropology is not an atheoretical practice. Martin is more hopeful on the integration of the interests of bioarchaeology and forensic anthropology, and has been involved in editing volumes that promote this integration of bioarchaeological and forensic anthropological approaches to research questions, e.g., [79,80]. She offers the term “forensic bioarchaeologist” as a means to promote this cross-disciplinary effort [78]. The term “forensic bioarchaeologist” was previously introduced by Scott and Connor [81], Skinner and colleagues [82], and Jessee and Skinner [83]. The latter two used it as a means of integration of archaeological methods and theory into medicolegal investigations of mass graves. Of particular pertinence to this discussion, Skinner et al. (2003) promoted guidelines for bioarchaeological practice in a forensic context. Nevertheless, the term “forensic bioarchaeologist”, has generally not been adopted.

Conversely, Steadman [84] discourages the use of such a term, arguing that it may serve to obscure the lines between forensic anthropology and bioarchaeology. In an academic sense, this may be “harmless” [84] p. 251; however, the term may cause confusion in the public about the distinction between the two disciplines, which she feels could potentially be problematic for jurors. We too argue the term could blur the boundaries of qualifications and expertise between the two disciplines, which is challenging for law enforcement and those working in the medicolegal realm. The further confounding of the two subdisciplines is also evidenced by the *American Journal of Physical Anthropology**’s* manuscript submission system. In this system, authors must designate a manuscript by “subfield”, with the only relevant choice for bioarchaeology or forensic anthropology being: “Bioarchaeology [including forensics]”.

We agree with Martin [78] p. 163, and Ubelaker [2] (as discussed previously) that the subdisciplines of bioarchaeology and forensic anthropology are independent and *complementary* and while they may differ in focus, contextual application, and specific hypotheses, they can learn much from each other. We firmly believe that a clear standardization of education, training, and qualifications is the best way for this mutual appreciation and professionalization to be achieved. The first step in this process is recognizing a lack of cross-disciplinary expertise, which can be achieved through a broader understanding of what constitutes expertise, as we discuss further below. 

## 2. Disciplinary Expertise

As we argued above and as others have demonstrated [74,85], while lacking published qualifications, bioarchaeology and forensic anthropology have developed into their own disciplines each with their own areas of expertise, bodies of literature, analytical methods, theoretical models, and education programs. However, it is important to discuss what expertise is and how it is created to fully appreciate the implications of differing expertise (and thus different disciplinary skills). Typically, we consider experts to have *authoritative* knowledge or skill in a particular area, while laymen are non-professional individuals lacking expertise [29,30,86,87,88]. A depth of literature has emerged relatively recently examining experience, expertise, and the sociology of scientific knowledge, e.g., [63,86,88,89,90,91,92,93,94,95,96,97,98,99,100,101,102,103,104,105,106,107]. We include a brief discussion of this literature here for some of the same reasons Collins [93] p. 127, was motivated to pioneer this avenue of inquiry, “to persuade sociologists [here, anthropologists] to reflect upon their expertises”.

Collins and Evans [108], present models of various forms of specialist expertise along a two- or three-dimensional spectrum [97,98,100]. Within specialist expertise exist two main types of knowledge, “Ubiquitous Tacit Knowledge” and “Specialist Tacit Knowledge” (Table 1). The first, “Ubiquitous Tacit Knowledge” (i.e., information) may be generated simply via reading without interacting with the appropriate contributory experts, this is knowledge that is easily accessible and therefore common knowledge. The novice level of “Ubiquitous Tacit Knowledge” is considered “beer mat” (knowledge of very superficial facts about a topic that you might find on a beer mat/coaster). The next level is “popular understanding”, which can be achieved through popular non-fiction books and general media. “Primary source knowledge” involves engaging with the primary literature; however, this literature still only provides “a shallow or misleading appreciation of science in deeply disputed areas”, which is far from obvious for the uninitiated [108] p. 22.

“Specialist Tacit Knowledge” must be acquired via interactions and enculturation with practicing professionals [100], and serves as the necessary knowledge base(s) to practice a discipline. Specialist Tacit Knowledge ranges from “interactional expertise” to “contributory expertise” [108]. Interactional expertise is essentially the ability to interact with other experts using their language/jargon and understanding the concepts being discussed, but lacking the full expertise to practice [102]. Contributory expertise is traditional technical expertise, where the practitioner is the contributory and interactional expert, meaning they are able to discuss/interact with other individuals at a complex level *and* able to perform complex disciplinary tasks competently [105]. With these definitions of types of knowledge, expertise can be defined as “the mastery of the tacit knowledge of a domain of practice, with interactional expertise being mastery of the domain’s language, and contributory expertise being the ability to competently engage in the practices of that domain” [104] p. 99.

As both bioarchaeology and forensic anthropology share many common lower-level knowledge areas, individuals educated in either discipline have some specialist knowledge of the other, representing what Collins and Evans [108] refer to as Primary Source Knowledge. For example, both may use the same method to estimate the sex of skeletal remains. However, as specialization increases, there is less and less overlap in knowledge, and the expertise required to interpret method results and generate reports becomes more exclusive. For example, bioarchaeologists must understand the historical context of the samples they are analyzing and when possible, work with descendent communities; while forensic anthropologists must understand jurisdiction, chain of custody, and admissibility of evidence.

These distinctions in knowledge area and specialist expertise are important, as without the appropriate “Specialist Tacit Knowledge”, practitioners/researchers may perform tasks inappropriately and/or incorrectly. As Collins and Evans [108] p. 22, state: “it can be shown that what is found in the literature, if read by someone with no contact with the core-groups of scientists who actually carry out the research in disputed areas, can give a false impression of the content of the science as well as the level of certainty”. In other contexts, this concept is often referred to as the Dunning–Kruger effect, or essentially the ignorance of one’s ignorance [109,110,111,112]. Individuals have just enough knowledge to understand the primary literature, but not enough to fully grasp the nuances of this material or how to properly discuss or apply it. The implications of which are that bioarchaeologists and forensic anthropologists, as contributory experts in their respective disciplines, can be ignorant of their lack of cross-disciplinary expertise. Collins and Evans state: “Enculturation” is the only way to master an expertise which is deeply laden with tacit knowledge because it is only through common practice with others that the rules that cannot be written down can come to be understood” [108] p. 24. Essentially, as knowledge becomes more specialized, individuals interested in acquiring this knowledge must rely on practitioners’ practices (i.e., experiential training programs) rather than literature (i.e., educational programs) [104]. Returning to bioarchaeology and forensic anthropology, essentially, the only way to develop contributory expertise in one of these disciplines is through enculturation in a bioarchaeological or forensic anthropological educational and/or training program supervised by a contributory expert in that discipline. This is not to say that individuals cannot be experts in both disciplines, rather it means that dual expertise requires individuals to develop contributory expertise in *both* bioarchaeology and forensic anthropology. As Collins and Evans [108] point out, lacking such enculturation at the level of contributory expertise leads to overconfidence and poor performance (i.e., the Dunning–Kruger effect). 

It is important to reiterate here that the focus is on education and training by other contributory experts, working towards building a body of knowledge and practical skills in a way that is consistent with how the discipline (i.e., other contributory experts) operates. This “enculturation” is not a form of limiting access to knowledge, but rather as means of acquiring knowledge in such a way that the learner will develop interactional expertise (being able to have high-level discussions with other contributory experts, using the appropriate processes) and contributory expertise (being able to use methods, technology, etc., in such a way that it contributes to the greater body of scholarly knowledge of a discipline). This is not a new concept and is essentially how academia currently operates. That is, students attend graduate school at programs that have education programs in which they are interested, working with advisors doing work similar to what they want to do as professionals. While academia is not without its major flaws, the argument here is simply that training and education are critical to gaining the requisite skills to perform disciplinary tasks. The arguments presented here are the first step in recognizing the need for developing expertise, the next step would be to develop such training and education programs. As a discipline, we can and should critically evaluate what this training looks like and how we define demonstrable expertise in a way that is inclusionary and equitable. 

## 3. Ethics, Expertise, and the Role of Professional Organizations 

Professional organizations like the American Academy of Forensic Sciences (AAFS), the American Association of Physical/Biological Anthropologists (AAPA/AABA), the Society for American Archaeology (SAA), and the American Anthropological Association (AAA) exist largely to provide individuals in that profession opportunities to network, organize, and serve and engage with the public. Additionally, these organizations typically provide professional development and continuing education opportunities, which is why student members are often encouraged to join as a means to facilitate disciplinary enculturation and entrée into the profession at large, which can also serve to provide them with the necessary expertise to practice their discipline. Further, organizational ethical codes should address qualifications so as to define the proper education and/or training to perform discipline-related tasks. Such a definition would prevent an individual from performing applied work outside their area of expertise, which is an ethical violation. Here, we first outline the need for professional ethical guidelines, and then we revisit the role these organizations can play in providing qualifications for members.

## 4. Why Do We Need Professional Ethical Guidelines? 

A professional is someone who: (1) possesses a body of special knowledge (i.e., contributory expertise), (2) practices within an ethical framework (i.e., adheres to a code of ethics and avoids conflicts of interest), and (3) fulfills a societal need [6,7,113]. Professionalism is conduct associated with a particular profession. Both bioarchaeology, e.g., [114,115,116,117,118,119] and forensic anthropology, e.g., [120,121,122,123,124] have extensive literature regarding ethical conduct and practice. However, ethical codes are typically established, and presumably enforced, by professional organizations. Ethical codes are used to: (1) establish conduct that is meant to be pursued by practitioners of a discipline (altruistic behavior); (2) establish conduct that must be avoided (i.e., misconduct), and (3) provide potential negative outcomes for practitioners engaging in misconduct [113,125]. Professional ethics are typically presented in the form of either aspirational guidelines or preventive standards [126]. Aspirational ethical codes are meant to promote human wellbeing and present a number of guiding and/or motivational behaviors that an organization *would like* its members to follow/achieve. Preventive ethical codes are enforced by an adjudicating committee within an organization that performs an investigation when a complaint of misconduct alleges that an individual acted unethically [113]. The content of ethical codes for professional organizations vary, but should generally be structured to ensure members avoid unprofessional conduct, so as to maintain the credibility of the profession and professional organization. Without clear professional ethics, a discipline does not have guidelines for acceptable or unacceptable behavior, such that there can be no good or bad conduct, and all actions must be treated equally [127] p. 233. 

In terms of the meaningful implementation of professional ethics, there are two essential issues that must be addressed. The first is that ethical codes must be detailed enough so that specific types of conduct considered to be unethical are demonstrably so. Second, ethical codes must be enforceable, with negative outcomes for individuals found guilty of misconduct. The importance of these issues is perhaps most easily demonstrable in terms of U.S. politics, where ethical guidelines are often ignored when ethical misconduct is not actually against the law, and the language of ethical guidelines is vague and not rigorously enforced [128,129,130]. As Josephson states: “there is a big difference between what you have the right to do and what is right to do” [131] p. 4. Unfortunately, the same is also the case in most professional organizations in which bioarchaeologists and forensic anthropologists are members. This is important as the law is meant to represent and enforce values for society as a whole, but is often not specific enough to cover many activities pertinent to a particular profession. Professional ethical codes more directly address discipline-specific values and behaviors.

Because ethical codes are tied to specific organizations, they only apply to the members of those organizations. This means that organizational membership (or lack thereof) plays an important role in establishing and policing the conduct of a profession and its body of practitioners, based on each organization’s ethical code. It also means that each organization should consider the ramifications of its membership requirements in terms of professional qualifications and access to students and non-experts, and how this allows the organization to serve its role within its professional community. Therefore, professional organizations serve a role of providing opportunities for gaining expertise through *enculturation* by interacting with additional contributory experts, and are positioned to provide sanctions when a member practices outside of their expertise, which could be seen as an ethical violation.

## 5. The Need for Disciplinary Qualifications

Bioarchaeology has no official or widely accepted published documents in the public or private sector regarding qualifications for bioarchaeological practice, or for the education or training of its practitioners. Currently, anyone *claiming* to have the appropriate training in bioarchaeology can be employed to perform applied bioarchaeological tasks. This is particularly true in contract archaeology (i.e., cultural resource management [CRM]), where it may be difficult to find qualified osteologists who are also available at the time of the excavation. These companies may then be forced to hire individuals with little osteological training to excavate and perform analyses.

As a recognition of the need for standardized qualifications, there have been some movements to define minimum qualifications to perform osteological analysis and excavation. Within the Code of Colorado Regulations, under Section 13 “Unmarked Human Graves”, point G states, “Pursuant to 24-80-1302(4)(e), the physical anthropological study of human remains shall be conducted by a qualified physical anthropologist with the credentials comparable to those required for principal investigators, as set forth in Section 5 of these regulations” (https://www.sos.state.co.us/CCR/GenerateRulePdf.do?ruleVersionId=541&fileName=8%20CCR%201504-7, accessed on 16 July 2021). The qualifications outlined for principal investigators include a graduate degree in anthropology, archaeology, or history with experience in Colorado archaeology; one year of professional experience; four months of supervised field and analytic experience; and the ability to complete research. 

The Wisconsin Historical Society has taken this a step further to establish specific guidelines to be a “qualified archaeologist for burial sites or a qualified skeletal analyst”. Their mandatory requirements include a graduate degree in anthropology, at least one year of professional experience or specialized training, at least four months of supervised experience, and the ability to complete a project. The full list of requirements and application instructions can be found on their website: https://www.wisconsinhistory.org/Records/Article/CS14963, accessed on 16 July 2021. The Society for California Archaeology (SCA) recently sent out an email to members with a draft outline for “recommended qualifications for field osteologists working in California”. Very generally, this guideline would recommend a master’s degree in anthropology and field experience dealing with human remains. There are additional qualifiers such as course work in human osteology, experience with NAGPRA, and a field school with an osteological focus, among others. The guidelines have not been published and are currently out for public comment (https://form.jotform.com/90855960158972, accessed on 16 July 2021). Of note, the guidelines would not be enforceable by the SCA, but would serve as recommendations for employers. These guidelines for qualifications address specialist expertise by requiring not only advanced education but also having already worked as a professional and having been supervised to gain enculturation.

Within forensic anthropology, Galloway and Simmons [132] identified deficiencies in the standardization of education and training in forensic anthropology over two decades ago. As a result, more formal efforts were taken up by the Scientific Working Group for Anthropology (SWGAnth) to establish guidelines for Qualifications [133], and Education and Training [134]. However, these documents were never widely adopted, nor are they enforceable. Passalacqua and Pilloud [85] surveyed practicing forensic anthropologists and found large variations in terms of graduate coursework taken by the survey participants. The survey also demonstrated a lack of consensus among practicing forensic anthropologists in what constituted appropriate education and training in forensic anthropology. However, there was overwhelming agreement that clear standards for education are needed, with a high degree of support (96%) for developing an accreditation for forensic anthropology educational programs. Additionally, Langley and Tersigni-Tarrant [135] outlined a model to develop qualifications in forensic anthropology based on medical education. In this model, there would be a set of core competencies demonstrated via various “entrustable professional activities”. Once core competencies are clearly identified and agreed upon, appropriate training and certification (to demonstrate expertise) could be implemented. 

There are currently no certifications for the profession of bioarchaeology, however, job ads in the United Kingdom for osteologists have added “professional grade membership of the CIFA [Chartered Institute for Archaeologists]” as a desired criterion; which is functionally a certification (albeit not necessarily focused on the analysis of human remains). 

While no widely supported guidelines or standards currently exist in terms of education and training, or qualifications within forensic anthropology, there are certification processes see, [7] for an overview. In Europe, the Forensic Anthropology Society of Europe (FASE) and the Royal Anthropological Institute (RAI) oversee certification processes. In Latin America, the Asociación Latinoamericana de Antroplogía Forense (ALAF) also has a certification process. In the United States, the ABFA has a certification process, and is currently the only accredited certifying body for forensic anthropology. 

While these certifications may exist, there is still a lack of clear qualifications (i.e., who can practice and how do you educate/train practitioners) within *both* subdisciplines. This lack of standardized qualifications is problematic as there are no widely agreed-upon standards for determining who is and who is not an expert and thus capable to perform tasks as a bioarchaeologist or a forensic anthropologist. As both disciplines are specialized, it can be difficult for outside agencies to accurately judge the qualifications and requisite expertise of individuals applying to perform these types of analysis. For example, with very few exceptions there are no standards in bioarchaeology to determine who can perform work for NAGPRA repatriation, osteological analysis, or excavation in CRM, or at academic archaeological sites. Nor are there requirements demonstrating expertise to teach bioarchaeological field-schools, or meaningful certifications or competencies gained through attending a field school. All of these things can be problematic for the adequate interpretation of archaeological sites and human remains with an irreplaceable loss of data and information (when analyses are permitted), particularly in cases of repatriation and reburial. Moreover, while there is a certification process within forensic anthropology, there is no legal requirement that a forensic anthropologist must be certified in order to perform such analyses in a medicolegal context. In fact, any such self-identified specialist can be tasked to perform this work. Contracting unqualified individuals to perform forensic anthropological casework can result in improper conclusions, which can hinder identification efforts (or worse, misidentify a person) and could have enormous consequences during the litigation process, for the analyst, their employer and, critically, for the family of the deceased.

The ramifications of the differences in the disciplines are that if an individual without the appropriate education and training acts beyond their professional expertise, they are misrepresenting their qualifications and could potentially do harm to the research project, field recovery, forensic case/investigation, descendant populations (as occurred with the well-known example of “The Ancient One”, also known as Kennewick Man [136,137,138], and/or the entire discipline; and are thus acting unethically. As such, professionals and professional organizations should be critically concerned about qualifications, expertise, and ethical practice. It cannot be expected that law enforcement agencies, attorneys, CRM firms, museums, or a medicolegal authority be trained in examining the nuances of an anthropological degree to determine who is and who is not qualified to be a bioarchaeologist or forensic anthropologist. There must be clear published standards that go beyond education in skeletal analysis as work in bioarchaeology and forensic anthropology has become increasingly specialized and individuated.

## 6. Conclusions: A Way Forward

We attempted to demonstrate that bioarchaeology and forensic anthropology have evolved and diverged into two separate disciplines, each encompassing its own suite of literature, education, training, and qualifications. Additionally, we attempted to illustrate the connection between expertise, ethics, and professional organizations as important aspects to the advancement of, and professional practice in, both bioarchaeology and forensic anthropology. Both disciplines are in need of the development of standardized education and training programs that reinforce best practice models for their applied foci. Once appropriate models for education and training have been defined, it can become possible for practitioners to demonstrate expertise to achieve credentials in a more meaningful and demonstrable way. Professional organizations should be leading these efforts in addition to establishing robust and enforceable ethical codes and tailoring their membership in such a way as to support the further professionalization of their disciplines. 

Thus far, bioarchaeology and forensic anthropology have not fully embraced standardization of practice or qualifications—although forensic anthropology is ahead in this regard [7]. This shortfall is probably due in part to the largely academic focus of both bioarchaeology and forensic anthropology and the lack of familiarity with program accreditation in anthropology generally (although this is common, if not required, in many other academic disciplines, including many of the sciences). However, the accreditation of academic programs specializing in bioarchaeology or forensic anthropology may be a relatively straightforward way to ensure the generation of expertise and qualifications. The generation of consensus-based qualifications (via graduate-level education) for these subdisciplines would not necessitate large changes in curricula within anthropology departments. Rather, these programs may need to make small adjustments to fit the required definitions for accreditation. For example, the definition of qualifications to be a bioarchaeologist could be graduate courses in osteology, bioarchaeology, archaeological theory, and a field school; courses that many bioarchaeologists would readily take and are currently offered by graduate programs. Further, this coursework could be modeled to allow for the development of competency of various related applied skills. For example, the osteology course could provide modules on human vs. non-human identification, or the field school could provide verification of expertise to adequately excavate and document skeletal material within an archaeological context. For forensic anthropology, graduate coursework with a forensic focus, osteology, and a forensic archaeological field school could be required. Competency could be demonstrated via mock or mentored casework. 

There are already movements to define qualifications within bioarchaeology and forensic anthropology; however, these should be codified by professional organizations and linked to education and training. Within bioarchaeology, there are regional movements to define qualifications to perform bioarchaeological work, but these efforts are in their earliest stages. As there is currently no professional organization for bioarchaeology, these steps are being taken by other organizations, such as the SCA, the Register of Professional Archaeologists (RPA), and the state governments in which the work is being performed. These qualifications are still vague and may not be broadly enforced or accepted by the professional community at large. It may be necessary for this work to be undertaken as a working group within the AAPA/AABA, or independently as is being done with forensic anthropology via the Organization of Scientific Area Committees for Forensic Science (OSAC) and the American Academy of Forensic Science Standards Board (ASB). Again, the definition of these qualifications would not serve as a means to hamper research or scientific integrity or as a means of keeping people out of the disciplines, but would aid in determining who is capable of performing disciplinary tasks in an applied setting. It may also be beneficial for bioarchaeology to develop a national certification process similar to the ABFA, which could serve to illustrate requisite expertise or competencies to employers and stakeholders. 

Within forensic anthropology, the OSAC and ASB are developing and have published best practice recommendations and standards for performing various disciplinary tasks. As part of this initiative, there is a consideration for education and training, and qualifications; however, these specific documents are not yet finalized. There is already a mechanism to review and approve education programs within the forensic sciences, the Forensic Science Education Programs Accreditation Commission (FEPAC). However, this organization does not currently oversee forensic anthropology programs. Still, the FEPAC model could provide a way forward for accrediting forensic anthropology (and bioarchaeology) educational programs, if necessary. Additionally, the ABFA certification process could be updated and improved to adequately illustrate competencies as outlined by the OSAC and ASB documents.

While this work is being undertaken to improve and standardize both disciplines, there is still very little consequence for not following existing standards or ethical codes. Platitudes on misconduct without operational enforcement mechanisms are not useful. As performing work beyond one’s qualifications is unethical, we argue that professional organizations need to more clearly define ethical codes with enforceable consequences. 

Bioarchaeology and forensic anthropology do not operate in a vacuum. Both disciplines examine the remains of deceased humans, and every action, use for, and result from that examination affects the beneficiaries and stakeholders associated with those human remains [139]. While not commonly considered, there has been a recent push to acknowledge that the dead retain their humanity and thus should be granted rights [113,140,141,142,143]. Additionally, human subjects have next-of-kin, be they direct living relatives, or more distantly related descendant-communities. As those performing these analyses are often responsible for the curation/custody of these remains, we must acknowledge that we have an ethical and moral responsibility to act in the best interests of these individuals and their next-of-kin (usually defined in open dialogue with relatives and kin). The analyses we perform are used to reach conclusions that are presented in reports, publications, and other media, available to not only the research subject’s next-of-kin, but also the public at large. Bioarchaeological reports may be used to understand past human lifeways and as one part of the process for repatriation and return to descendent communities. Forensic anthropology reports may be used not only to bring closure to a family, but in court to adjudicate innocence or guilt. The conclusions of our work have meaning and reflect upon the identity and lives of the deceased as well as the communities from which they came. 

When we consider the importance of this type of work, we should want to ensure that the individual performing an analysis is an expert, and we owe that commitment to our communities and the individuals we study. Incorrect analyses and erroneous conclusions cause harm. As such, we should want to ensure that our students are being educated in the methods and skills necessary to perform their work and best serve not just their discipline, but their research subjects. Further, we should want to ensure that the quality of work we perform as professionals continues to be held to a high standard by generating and following best practices and/or standards for analytical decisions in applied practice. We should do these things, not just because they are the right thing to do, but as a service to those upon which we base not only our work, but also our entire careers.

## Figures and Tables

**Table 1 biology-10-00691-t001:** Levels of expertise based on Collins and Evans (2007).

SPECIALIST EXPERTISES	UBIQUITOUS TACIT KNOWLEDGE			SPECIALIST TACIT KNOWLEDGE	
	Knowledge That Is Easily Accessible (i.e., Ubiquitous)			Exclusive Knowledge That Must Be Acquired via Interactions and Enculturation with Practicing Contributory Experts	
	Beer Mat Knowledge	Popular Understanding	Primary Source Knowledge	Interactional Expertise	Contributory Expertise
	Knowledge of very superficial facts about a topic such that you might find on a beer mat/coaster	Knowledge based on popular non-fiction books and the general media	Knowledge based on engaging with the primary literature.	This represents having enough expertise about a discipline to interact with its contributory experts performing their work, but lacking the technical knowledge to perform it yourself.	This represents having enough expertise to contribute to a discipline through its technical and scholarly practice
			Note that literature still only provides “a shallow or misleading appreciation of science in deeply disputed areas” (Collins and Evans 2007:22)	“Scientists themselves tend to have contributory expertise in their narrow specialism and interactional expertise in cognate specialisms”. (Collins 2004:141)	

## Data Availability

Not applicable.

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
