# Peer review of "Forensic Anthropology as a Discipline"

_biology, 2021, doi:10.3390/biology10080691_

Round 1

Reviewer 1 Report

This manuscript is a survey of the forensic anthropology as a specialization within the overall field of physical anthropology. After an introduction which deals whith the different goals of bioarchaeology and forensic anthropology, the authors demonstrate that both disciplines have evolved and diverged into two separate disciplines, as well as illustrates the issues of expertise, training and ethical guidelines necessary to search, recover, and examine modern human skeletal remains within a medicolegal context. Althought the disciplines of bioarchaeology and forensic anthropology vary greatly globally (in terms of education and practice), the authors focus the manuscript on the practice of these disciplines within USA.

Overall, the mansucript is very clear, well-structured and well referenced.

Comments:

  • I believe that a discussion about the roles of forensic anthropologists/bioarchaeologists in mass graves in terms of practical versus management/planning/supervisory roles would be interesting in this manuscript. In its current form, there is no mention about how forensic anthropology is practiced in mass grave contexts, but increasingly I believe we are seeing forensic anthropologists take more important roles in these contexts, managing the overall Project and working with bioarchaeologists to facilitate the overall resolution of the mass grave research. I think this should be more fully addressed.
  • Table 1 is cited in the text but is not included in the proof for peer review.

Author Response

I believe that a discussion about the roles of forensic anthropologists/bioarchaeologists in mass graves in terms of practical versus management/planning/supervisory roles would be interesting in this manuscript. In its current form, there is no mention about how forensic anthropology is practiced in mass grave contexts, but increasingly I believe we are seeing forensic anthropologists take more important roles in these contexts, managing the overall Project and working with bioarchaeologists to facilitate the overall resolution of the mass grave research. I think this should be more fully addressed.

  • We thank the reviewer for their comments. We agree that a discussion of mass graves and the different roles that forensic anthropologists play in their investigation is interesting. Nevertheless, due to the varied practice and definition of bioarchaeologists and forensic anthropologists (also bioarchaeologists) in different countries and contexts, we have decided to limit our focus to the professionalization of the disciplines, rather than the activities that these professionals perform. We believe that eventually this discussion will extend to examine particular practices at a national, regional, international level, but we also feel it necessary to first establish clarity about how to define disciplinary expertise before broadening the discussion.

Table 1 is cited in the text but is not included in the proof for peer review.

  • This may be due to a formatting issue – we hope this is resolved in the resubmission and the table is visible now

Reviewer 2 Report

Review of the manuscript Forensic anthropology as a discipline.

The authors presented an article written in understandable language, although with rather complex writing style.

Unfortunately, the article does not offer a balanced snapshot of the discipline. The forensic anthropology practice around the globe were not accounted for. The authors’ statement "Arguably, the first journal dedicated to bioarcheology was the International Journal of Osteoarchaeology, [...]. Although, like the term osteoarchaeology itself, this journal has a heavy European focus [...]" (p. 4, Paragraph 3), I allow myself to state the article has heavy North American focus. Although stated that "While bioarcheology and forensic anthropology can vary greatly in their education and practice globally, we focus on the practice of these subdisciplines primarily within the United States, although we highlight other countries and their development in these efforts when relevant.", almost exclusively American Societies/Guidelines were recalled. The relevance of the discipline in Europe and Latin America was 'gratified' with one sentence in each case (p. 9, Para 3, as far as I could comprehend), while the bioarcheology and forensic anthropology in other regions/countries (India, Australia, Southeast Asia, etc.) were not considered as relevant. There are many articles about forensic anthropology practice worldwide, which were not referenced (e.g. Kranioti and Paine (2011) Forensic anthropology in Europe: an assessment of current status and application, Obertova et al. (2019) The Status of Forensic Anthropology in Europe and South Africa: Results of the 2016 FASE Questionnaire on Forensic Anthropology, Traithepchanapai et al. 2016 History, research and practice of forensic anthropology in Thailand, Go 2018 Appraising forensic anthropology in the Philippines: Current status and future directions, Jankauskas 2009 Forensic anthropology and mortuary archaeology in Lithuania, Vaswani and Ahmed (2020) Forensic anthropology education and training in India).

There are large differences in the educational background of forensic anthropologist across the globe. As Prof. Brinkmann wrote, "This is an important difference to the situation in the USA where forensic anthropology is only practiced by anthropologists. Another difference is that forensic anthropologists in Europe do not only examine bones." (Brinkmann 2007 - Forensic anthropology (https://pubmed.ncbi.nlm.nih.gov/29574835/)). For example, there are just few similarities in the education in the USA and Germany. I find it as unusual, that the authors almost completely neglected the fact that a substantial number of practicing forensic anthropologist have medical background. For example, the authors wrote in the introducing sentence of the Abstract "Forensic anthropology is a specialization within the overall field of anthropology. [...] Over time forensic anthropology has become increasing specialized and distinct from other specializations within anthropology.". Hence, excluding the possibility of educational trajectories outside the anthropology might lead to expertise in forensic anthropology". On the contrary, "Forensic anthropology shall be defined as the application of the science of physical anthropology and archaeology to the legal process". [American Board of Forensic Anthropology, Inc. Policies and Procedures Manual.].

Moreover, since the assessment of bone findings enables the investigating authorities to classify the (criminal) legal relevance, the judiciary systems worldwide (may) demand deployment of experts with standardized education and training program, and legally recognized professional organizations (e.g. Chambers).

I agree with the authors that the forensic anthropology as a discipline may be "endangered" by self-identified forensic anthropologist and bioarcheologist, and that there is a need for disciplinary qualifications. Unfortunately, I am not convinced that the article in the current form brings us closer to the solution of the problem. Moreover, "The divergence of bioarchaeology and forensic anthropology" (subtitle 1.2.) is not (easily) distinctive from the text.

The  authors quoted their unpublished work at least 13 times (Reference 7 - In Press, e.g., “see, 7 for an overview” (p. 9, para 3))—however, this paper is publicly not available.

Several segments of the text are fuzzy and hard to follow. To the majority of audience is English not a mother tongue and to the reviewer either. As Prof. Sainani from Stanford University said - the Science itself is complex enough - there is no need to use a complicated language. Hence, the authors should consider "keeping it simple": simple expressions, common words, avoiding 'fancy' language.

Furthermore, the Table 1 (p. 5, para 4) was not presented. Also, I am not sure what the numbers in the brackets next to many references mean—e.g., [63:251]. 

Author Response

The authors presented an article written in understandable language, although with rather complex writing style.

Unfortunately, the article does not offer a balanced snapshot of the discipline. The forensic anthropology practice around the globe were not accounted for. The authors’ statement "Arguably, the first journal dedicated to bioarcheology was the International Journal of Osteoarchaeology, [...]. Although, like the term osteoarchaeology itself, this journal has a heavy European focus [...]" (p. 4, Paragraph 3), I allow myself to state the article has heavy North American focus. Although stated that "While bioarcheology and forensic anthropology can vary greatly in their education and practice globally, we focus on the practice of these subdisciplines primarily within the United States, although we highlight other countries and their development in these efforts when relevant.", almost exclusively American Societies/Guidelines were recalled. The relevance of the discipline in Europe and Latin America was 'gratified' with one sentence in each case (p. 9, Para 3, as far as I could comprehend), while the bioarcheology and forensic anthropology in other regions/countries (India, Australia, Southeast Asia, etc.) were not considered as relevant. There are many articles about forensic anthropology practice worldwide, which were not referenced (e.g. Kranioti and Paine (2011) Forensic anthropology in Europe: an assessment of current status and application, Obertova et al. (2019) The Status of Forensic Anthropology in Europe and South Africa: Results of the 2016 FASE Questionnaire on Forensic Anthropology, Traithepchanapai et al. 2016 History, research and practice of forensic anthropology in Thailand, Go 2018 Appraising forensic anthropology in the Philippines: Current status and future directions, Jankauskas 2009 Forensic anthropology and mortuary archaeology in Lithuania, Vaswani and Ahmed (2020) Forensic anthropology education and training in India).

  • We thank reviewer two for their comments. Perhaps we were not sufficiently clear in defining the limited scope of our article. We are responding to debates in the United States about who can/should perform forensic anthropological and bioarchaeological work and how that is, or ought to be, established. We recognize that this has implications for other countries, but our starting point is a single context – the United States; and we have revised the text slightly to make this clearer. The authors have worked in more than 25 countries, so we are very aware that education, professional standardization, and practice are all distinct from the US and for this reason, our focus is strictly on the disciplines (forensic anthropology and bioarchaeology) in the US. We do this because we believe that it would be too complex to propose universal standardization/definition, when even national standardization does not yet exist.
  • We also added many citations on the state of forensic anthropology in other countries across the globe

There are large differences in the educational background of forensic anthropologist across the globe. As Prof. Brinkmann wrote, "This is an important difference to the situation in the USA where forensic anthropology is only practiced by anthropologists. Another difference is that forensic anthropologists in Europe do not only examine bones." (Brinkmann 2007 - Forensic anthropology (https://pubmed.ncbi.nlm.nih.gov/29574835/)). For example, there are just few similarities in the education in the USA and Germany. I find it as unusual, that the authors almost completely neglected the fact that a substantial number of practicing forensic anthropologist have medical background. For example, the authors wrote in the introducing sentence of the Abstract "Forensic anthropology is a specialization within the overall field of anthropology. [...] Over time forensic anthropology has become increasing specialized and distinct from other specializations within anthropology.". Hence, excluding the possibility of educational trajectories outside the anthropology might lead to expertise in forensic anthropology". On the contrary, "Forensic anthropology shall be defined as the application of the science of physical anthropology and archaeology to the legal process". [American Board of Forensic Anthropology, Inc. Policies and Procedures Manual.]. Moreover, since the assessment of bone findings enables the investigating authorities to classify the (criminal) legal relevance, the judiciary systems worldwide (may) demand deployment of experts with standardized education and training program, and legally recognized professional organizations (e.g. Chambers).

  • In the same manner, due to this narrow focus, we do not consider other educational routes to professional forensic anthropological or bioarchaeological practice because this does not reflect current practice in the United States.

I agree with the authors that the forensic anthropology as a discipline may be "endangered" by self-identified forensic anthropologist and bioarcheologist, and that there is a need for disciplinary qualifications. Unfortunately, I am not convinced that the article in the current form brings us closer to the solution of the problem. Moreover, "The divergence of bioarchaeology and forensic anthropology" (subtitle 1.2.) is not (easily) distinctive from the text.

  • We hope that with these revisions, the reviewer sees the value in this manuscript
  • Hopefully in the published version the subheading will be more readily noticeable, as this is the formatting of the journal

The authors quoted their unpublished work at least 13 times (Reference 7 - In Press, e.g., “see, 7 for an overview” (p. 9, para 3))—however, this paper is publicly not available.

  • This paper has since been published and is available for review. We have updated the references to reflect this recent publication.

Several segments of the text are fuzzy and hard to follow. To the majority of audience is English not a mother tongue and to the reviewer either. As Prof. Sainani from Stanford University said - the Science itself is complex enough - there is no need to use a complicated language. Hence, the authors should consider "keeping it simple": simple expressions, common words, avoiding 'fancy' language.

  • We have reviewed the manuscript for language and revised several words to select more common synonyms so that it will be more easily understood

Furthermore, the Table 1 (p. 5, para 4) was not presented. Also, I am not sure what the numbers in the brackets next to many references mean—e.g., [63:251]. 

  • We hope this is resolved in the resubmission and Table 1 is visible
  • The numbers next to the brackets are the page numbers for the references, as per the journal guidelines

Reviewer 3 Report

Thank you for offering me the opportunity to review this article. I really enjoyed reading it. 

This paper exposes the current state of forensic anthropology and the need to outline requisite qualifications, develop standards and best practice guidelines, and enforce robust preventive ethical codes in order to define forensic anthropology as a discipline of its own. Furthermore, they attempt to show that bioarchaeology and forensic anthropology
have evolved and diverged into two separate disciplines.

It is an issue that is widely debated in forensic anthropology congresses, but there are few articles dealing with it, hence the interest for the scientific community.

However, I would like to make a few comments and recommendations: 

1. I have missed a comparison of the state of forensic anthropology and bioarchaeology in different countries.

2. Please,  check for typos especially in the Introduction section.

Author Response

Thank you for offering me the opportunity to review this article. I really enjoyed reading it.   This paper exposes the current state of forensic anthropology and the need to outline requisite qualifications, develop standards and best practice guidelines, and enforce robust preventive ethical codes in order to define forensic anthropology as a discipline of its own. Furthermore, they attempt to show that bioarchaeology and forensic anthropology
have evolved and diverged into two separate disciplines.  It is an issue that is widely debated in forensic anthropology congresses, but there are few articles dealing with it, hence the interest for the scientific community.

However, I would like to make a few comments and recommendations: 

  1. I have missed a comparison of the state of forensic anthropology and bioarchaeology in different countries.
  • We thank the reviewer for their comments. Due to the varied practice and definition of forensic anthropologists (also bioarchaeologists) in different countries and contexts, we have decided to limit our focus to the United States. We believe that extending this discussion to examine practice at a regional, international level is interesting and important, but we also feel it necessary to first establish clarity in a single context, before broadening the debate. We have revised the text slightly to make this clear.

  1. Please,  check for typos especially in the Introduction section.
  • We have revised the text for typos and made pertinent changes.

Round 2

Reviewer 2 Report

I would like to thank the authors for the clarifications and for revising the article. Since the article relates only to the forensic anthropology in the United States, I would suggest to state that in the title.

Unfortunately, the authors failed to present the “current state of forensic anthropology in the United States as a distinct discipline.”, neither “how forensic anthropology has diverged from other specializations of anthropology enough to be considered its own discipline” nor they have “clearly defined qualifications, training, standards of practice, certification processes, and ethical guidelines”, as announced in the abstract.